# A Fully Integrated High Efficiency 2.4 GHz CMOS Power Amplifier with Mode Switching Scheme for WLAN Applications

**Haoyu Shen** [1,2,*]**, Taishan Mo** [3] **and Bin Wu** [3,*]

1   The Intelligent Manufacturing Electronics R&D Center, Institute of Microelectronics of the Chinese Academy of Sciences, Beijing 100029, China
2   School of Electronic, Electrical and Communication Engineering, University of Chinese Academy of Sciences, Beijing 100049, China
3   Zhejiang CASEMIC Electronics Technology Co., Ltd., Hangzhou 310051, China; motaishan@casemic.com
*   Correspondence: shenhaoyu@ime.ac.cn (H.S.); wubin@casemic.com (B.W.)

**Abstract:** A 3.3 V mode-switching RF CMOS power amplifier (PA) for WLAN applications is presented, which is integrated into a 55-nm bulk CMOS process. The proposed PA offers both static control and dynamic power control, allowing it to operate efficiently in both low-power and high-power modes. The pure low-power mode is achieved by reducing power cells, which are also used for linearization in high power mode. The low-power mode is achieved by reducing the number of power cells which are also used for linearization in the high-power mode. In the dynamic power control mode, the total AM–AM and AM–PM distortion is effectively compensated for by dynamically controlling the number of power cells and adjusting the matching input. The proposed PA achieves an output P1dB of 27.6 dBm with a PAE of 32.7% and an output P1dB of 17.7 dBm with a PAE of 10% in high-power and low-power modes, respectively. It is measured with an 802.11 n 64-quadrature-amplitude-modulation (MCS7) signal and shows a maximum average power of 19 dBm under an error-vector-magnitude (EVM) of −27 dB.

**Keywords:** CMOS power amplifier (PA); dual mode; power control; wireless local network (WLAN)

## 1. Introduction

Wireless local area network (WLAN) technology has become increasingly popular in recent years, and has been integrated into various mobile devices such as computers, mobile phones, and tablets. Research on CMOS SOC that can be used in wireless local area network (WLAN) technology has been the focus of attention in recent years [1–5]. However, integrating power amplifiers (PAs) into WLAN systems has been a major bottleneck due to the low breakdown voltage and the lossy substrate in the CMOS process.

Due to the lossy substrate in the CMOS process, the low-Q on-chip inductors and transformers seriously affect the linearity and efficiency of the PA. Off-chip matching can be used to solve these problems [6], but it increases the complexity and cost of the system. Moreover, directly connecting the drain of the transistor to the output PAD can cause severe breakdown problems that need to be carefully solved using ESD protection circuits without affecting linearity. Preferably, fully integrated PAs offer a better solution to help WLAN SOC achieve better gain control and system calibration, such as digital pre-distortion (DPD) loopback and adjustable gain control (AGC).

Other roadblocks to the integration of high-performance power amplifiers (PAs) are low breakdown voltage and low current driving capabilities. To address these challenges, various techniques have been employed in CMOS PAs, such as series-combing transformers/parallel-combing transformers (SCTs/PCTs) [7,8] and multistage cascode topology, aiming to enhance output power and gain. However, the SCTs and PCTs require extensive layout design to mitigate additional phase shifts and coupling between transformers, which can adversely impact the linearity and efficiency of the PA due to

their large areas. Similarly, the multistage cascode topology has its own limitations attributable to the common gate (CG) transistor, as the presence of extra non-linearities from the CG transistor can degrade the overall linearity performance of the PA, including the bias circuit impedance of the CG transistor. Reference [6] indicates that the gate bias circuit of the CG transistor can introduce additional IMD3 asymmetry, which constitutes a critical factor in determining the linearity of the PA. Eliminating this non-linear effect adds complexity to the design. Nonetheless, despite these challenges, CMOS PAs remain a focal point of research due to their low cost and high integration capabilities. In particular, multi-band, multi-standard transceivers can be integrated into one single chip [9–11], which greatly reduces off-chip components, thereby reducing both costs and the PCB area. Therefore, CMOS SOC are widely used in portable devices such as mobile phones, watches, and tablets.

The modern Wireless Local Area Network (WLAN) 802.11ax/ac standard supports high data throughputs with strict linearity requirements, including support for modulation schemes such as 1024 and 256 quadrature amplitude modulation (QAM) and bandwidths up to 160 MHz. The linearity requirements are primarily characterized by two main factors: the error vector magnitude (EVM) and the spectral emission mask. In the case of transmitters, the linearity of the power amplifier (PA) plays a crucial role in determining the overall linearity of the entire transmitter. However, when compared to the PA design of III–V compound technology [12–14], CMOS PAs exhibit relatively poorer linearity. Consequently, various linearization techniques have been employed to enhance the linearity of CMOS PAs and meet the EVM requirements of WLAN modulation signals. These techniques include the employment of MGTR (Multigated Transistors) [15,16], analog and digital pre-distortion technology [17–19], feedback and feedforward technology [20,21], and adaptive bias circuits [22]. The adoption of these techniques aims to mitigate linearity limitations and improve the overall performance of CMOS PAs so they can meet the stringent requirements of WLAN modulation signals. Additionally, WLAN standards support MIMO technology, which allows users to use different spatial streams to improve throughputs. WLAN also supports multiple bands, so it is necessary to design antennas [23–25] with compact, weak coupling and high radiation efficiency performance to achieve high signal efficiency and high linearity transfer.

Moreover, WLAN uses orthogonal frequency division modulation (OFDM) technology to support higher throughputs, which lead to a high peak-to-average ratio (PAPR). Therefore, PAs need to have high efficiency both at peak output power and at large back-off output power in order to save power. To improve efficiency at the power back-off level, various technologies have been developed in recent years. Supply envelope tracking is an effective method [26–28], but in the CMOS process, the power supply voltage is relatively low, which makes it difficult to apply to CMOS PAs; the Doherty PA is another popular way to improve back-off power efficiency [29–32], but there are also difficulties regarding auxiliary PA matching and layout routing. Recently, some new technologies have been used to further improve PA efficiency at deep PBO levels. Multi-ways Doherty PAs can achieve multiple efficiency peaks at PBOs with a relatively larger size [33–36]; references [37,38] propose a class-G Doherty PA, which achieves efficiency peaks at 6- and 12-dB PBOs, with additional hardware overhead.

In this article, we propose a fully integrated linear CMOS PA operating from a 3.3-V supply for 2.4 G WLAN applications. It not only achieves high linearity using gm linearization, PMOS compensation, and large signal MGTR techniques [39], but also has a high efficiency at high back-off power with the proposed method. This article is organized as follows: In Chapter 2, the efficiency-enhanced method is discussed. The detailed circuit implementation is described in Chapter 3. Chapter 4 presents the measurement results, Chapter 5 discusses the future research direction of CMOS Pas for WLAN, and Chapter 6 concludes this article.

## 2. Efficiency-Enhanced Method

To address the issue of low power back-off efficiency caused by the high peak-to-average power ratio (PAPR) of WLAN signals, this work proposes a fully integrated PA with two operation modes to improve back-off efficiency. The PA consists of two stages, and the power cells of the second stage are reconfigurable and divided into five units. As illustrated in Figure 1, the proposed PA has two operation modes: high-power mode and low-power mode. In high-power mode, all power cells in the second stage of the PA are activated for high output power. Additionally, gm linearization, PMOS compensation, and the on-chip high Q output matching network are employed to ensure high linearity and output power. In the low-power mode, some of power cells in the second stage of the PA are turned off to improve efficiency and reduce DC power consumption, because the second stage of the PA mainly affects the total efficiency. The power-added efficiency (PAE) can be improved by × 1.5 in low-power mode with the same output power in the high-power mode. Moreover, with the on-chip power controller circuit dynamically configuring the PA's operation mode according to the output power level, the PA allows not only static control but also dynamic control. In the dynamic control switching mode, the average power-added efficiency (PAE) over the entire output power range is 1.5 times higher than that of the PA without mode switching.

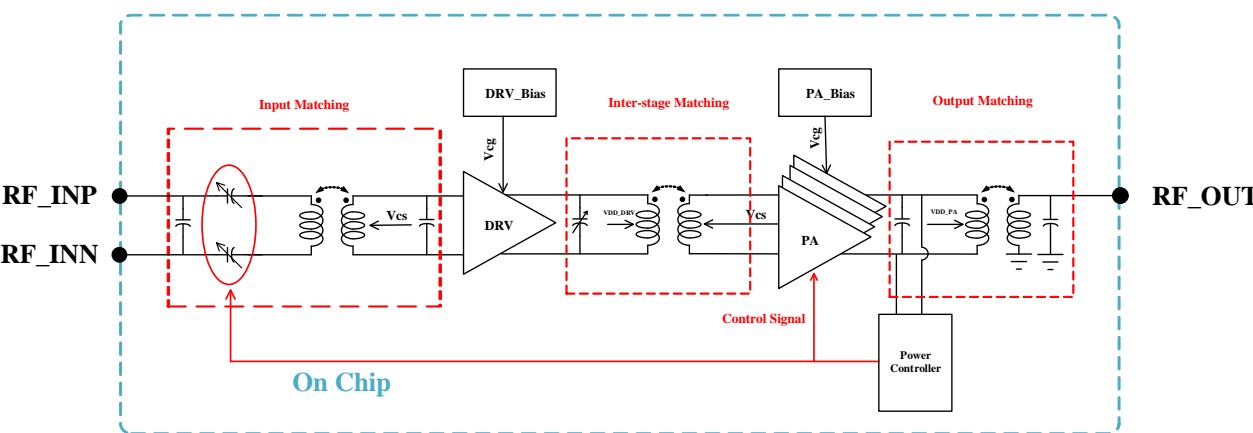

**Figure 1.** Architecture of the proposed CMOS PA.

In the case of dynamic mode switching, the power controller detects the power of the circuit and generates a control signal to adjust the power cells and input capacitors. When operating at low-output power levels, the power detector detects that the power is low and generates a control signal to turn off four of the five power amplifier units, leaving only one unit turned on. As a result, the power-added efficiency (PAE) can be improved due to the reduced power cell size, and the DC current can also decrease significantly. As the output power continues to increase, the remaining four units are turned on by the control signal. In high power conditions, all power cells are turned on to provide high linearity and power. Through the method of this work, the back-off power efficiency can be greatly improved.

Figure 2a shows the simplified single-ended schematic of the proposed reconfigurable power cells. The small signal equivalent circuit of the power cell is shown in Figure 2b. Assuming that all the transistors work in the saturation region and the output impedance is ignored, the small signal voltage gain can be derived as

$$A_v = \frac{V_{out}}{V_{in}} = -Z_L \frac{jwM}{(Rs + R_L) + (jwLs + jX_L)} \cdot gm2 \left\{ gm1 - \frac{jwC1g_{m1}}{jwC1 + gm2\frac{C1}{Cgs2}} \right\}$$

where $Z_L = R_L + jX_L$, $M = k\sqrt{L_1 L_2}$. Thus, there will be phase and amplitude discontinuity during the dynamic mode switching between the high-power mode and low-power mode, causing AM–AM and AM–PM distortion.

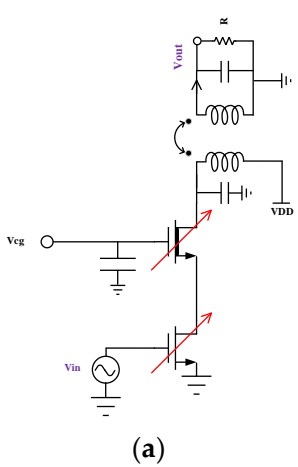

**(a)**

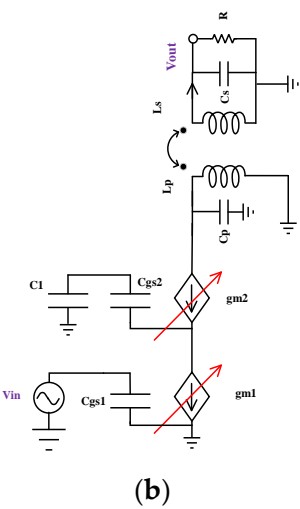

**(b)**

**Figure 2.** (**a**) Simplified schematic of the proposed reconfigurable single-ended power cells. (**b**) The small signal equivalent circuit.

The large AM–AM and AM–PM distortion may cause severe linearity degradation. Reference [40] shows that the ACLR degrades rapidly with larger gain and phase discontinuities. To address the issue of gain and phase discontinuities caused by the dynamic power control mode, this work proposes the addition of variable capacitors at the input matching to ensure minimal discontinuities in total gain and phase.

Therefore, in order to achieve high performance, careful design considerations must be taken into account. Specifically, the gain variation at the input matching must be less than that of the second stage PA, and the total variation of the gain should not exceed 1~1.5 dB. This approach can help minimize distortion and ensure high linearity. The remaining distortions could be compensated by digital pre-distortion (DPD).

Power cells are carefully designed to maximize efficiency and linearity. The detailed circuit design will be presented in Chapter 3.

## 3. Circuit Implementation

As shown in Figure 1, the proposed power amplifier (PA) is composed of a power control circuit, two-stage power cells, on-chip matching circuits(including on-chip transformers and capacitors),and integrated bias circuits. The power cells utilize a differential structure to compensate for even-order harmonics and common-mode fluctuations caused by ground bonding wires.

### 3.1. Power Controller Circuit

Figure 3 depicts the schematic of the power controller circuit. The power controller circuit consists of a power detector and comparators. The power detector detects the RF power signal and outputs a DC voltage, which enters the comparator with a given reference voltage to generate control signals for the power cells and capacitors.

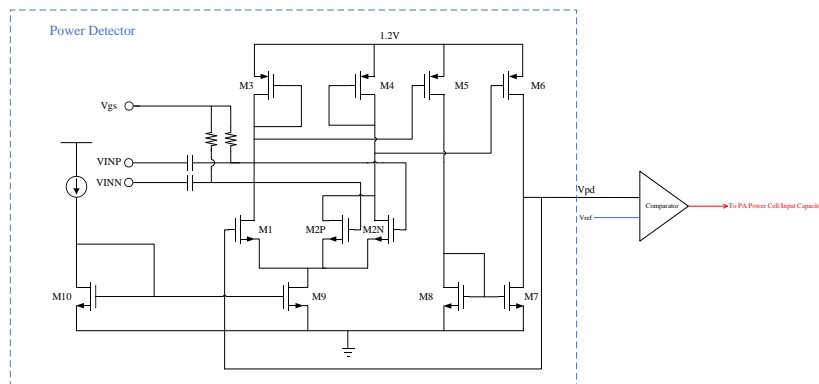

**Figure 3.** Schematic of the Power Controller.

The power detector is composed of a two-stage amplifier. The RF differential signals enter M2P and M2N, whose W/L is half of M1. For small signals, the total current of the M2 branch does not change due to the differential inputs, and the output voltage Vpd is the same as Vgs due to the feedback. As the power increases, the total current of the M2 branch increases. Due to the tail current source, M9, the drain voltage of M2 increases, and the output DC voltage Vpd increases accordingly.

The gain-bandwidth product (GBW) of the power detection circuit is 200 MHz and meticulously designed to suppress the carrier while ensuring minimal impact on the signal bandwidth of the wireless local area network (WLAN). Then, the output signal is directly compared with a predetermined reference voltage. Whenever the output voltage of the power detection circuit is higher than the reference voltage, the comparator generates a high-level signal, consequently activating the entire circuit to operate in a high-power mode. Conversely, if the output voltage falls below the reference voltage, the circuit switches to a low-power mode. The reference voltage is adjustable, allowing for the generation of control signals at different power levels.

The overall time delay of the power controller circuits, from the moment the input signal enters the circuit until the digital control signal is generated, is approximately 3 ns. This time delay effectively satisfies the signal output requirements of the WLAN. Moreover, the circuit maintains a low current consumption level under 1.2 V supply voltage, thus minimizing its impact on the overall efficiency of the power amplifier.

Figure 4 illustrates the simulated power detector voltage versus input power, which demonstrates good linearity over the entire power range.

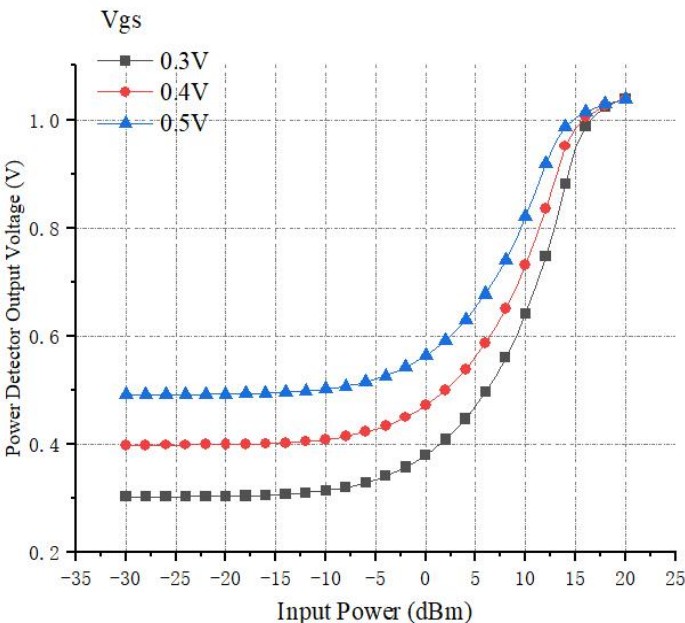

**Figure 4.** Simulated power detector output voltage.

Then, the comparison between the output detector voltage Vpd and different adjustable reference voltages generates control signals. Figure 5 shows the simulated logic control voltages versus input power.

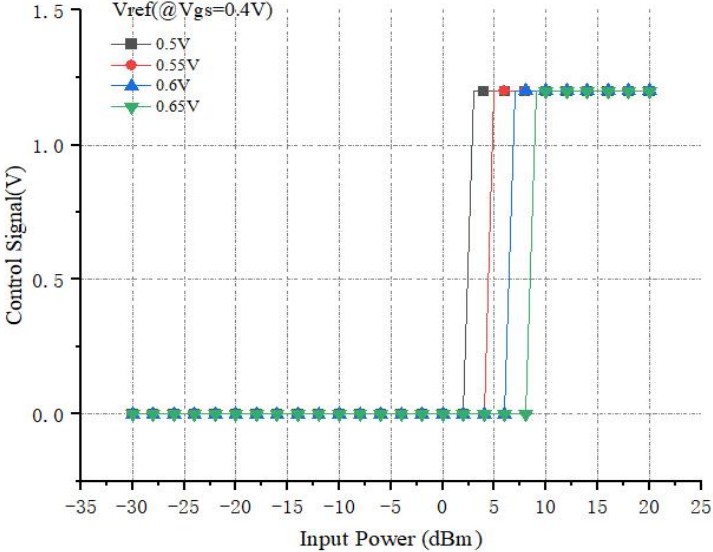

**Figure 5.** Simulated output control signal.

### 3.2. Linearity Enhanced Method

Figure 6 shows the simplified schematic of power cells and Table 1 shows the power cell transistor size. To ensure the flexibility of power control, the power cells are designed to be reconfigurable and divided into five units. Each cell uses a cascode structure and can be individually biased, which enables dynamic control by the power controller circuit as well as static control. In the common gate-transistors, thick oxide is used for high supply voltage, while thin oxide is used in the common-source transistors to provide high gain. Additionally, a parallel PMOS at the input is added to improve linearity.

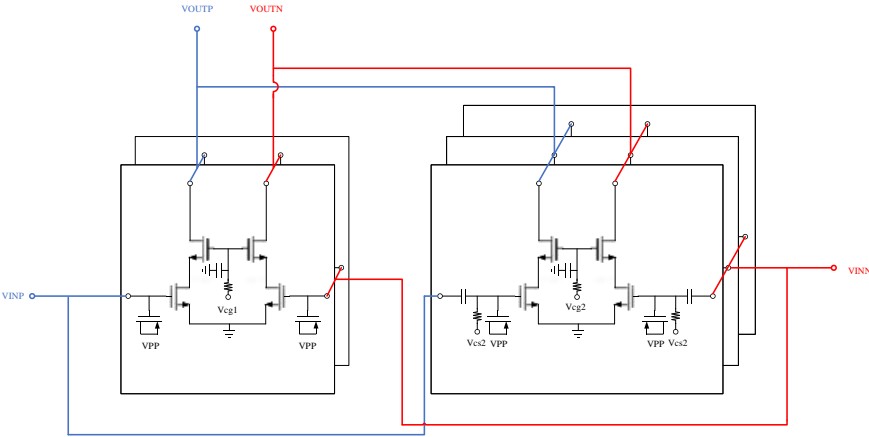

**Figure 6.** Simplified schematic of the power cells.

**Table 1.** Power Cell Transistor Size.

|  | W/L |
| --- | --- |
| Common Source | 800 μm × 2/100 n |
| Common Gate | 800 μm × 2/500 n |

In general, power amplifiers (PAs) are biased to Class AB for a compromise between linearity and efficiency. In this design, the second stage core devices are divided into five groups that are biased to different voltages. When all the groups are turned on, some groups are biased to Class AB but closer to Class A, while others are biased to Class AB but closer to Class B. The ratio of these two groups is 2:3, which achieves a relatively flat gm over a wider range of input voltage, thus obtaining better linearity at the expense of small signal voltage gain. Figure 7 illustrates the overall linearity, which is greatly improved.

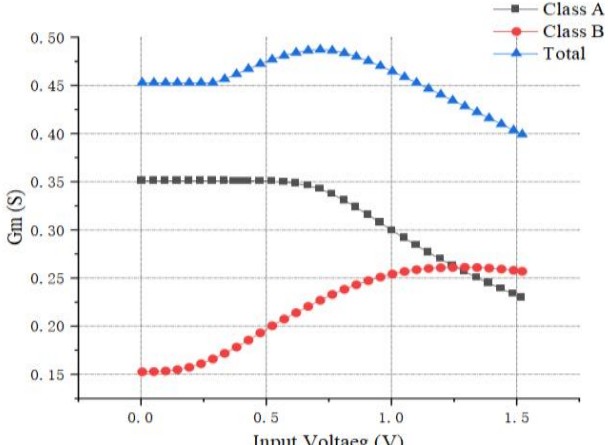

**Figure 7.** Simulated Gm for Class A and Class B devices, and overall Gm.

Figure 8 shows the simulated comparison of the output power between with- and without-gm linearization. The results show that with the help of gm linearization, the PA has a higher OP1dB compression point, indicating a higher linearity.

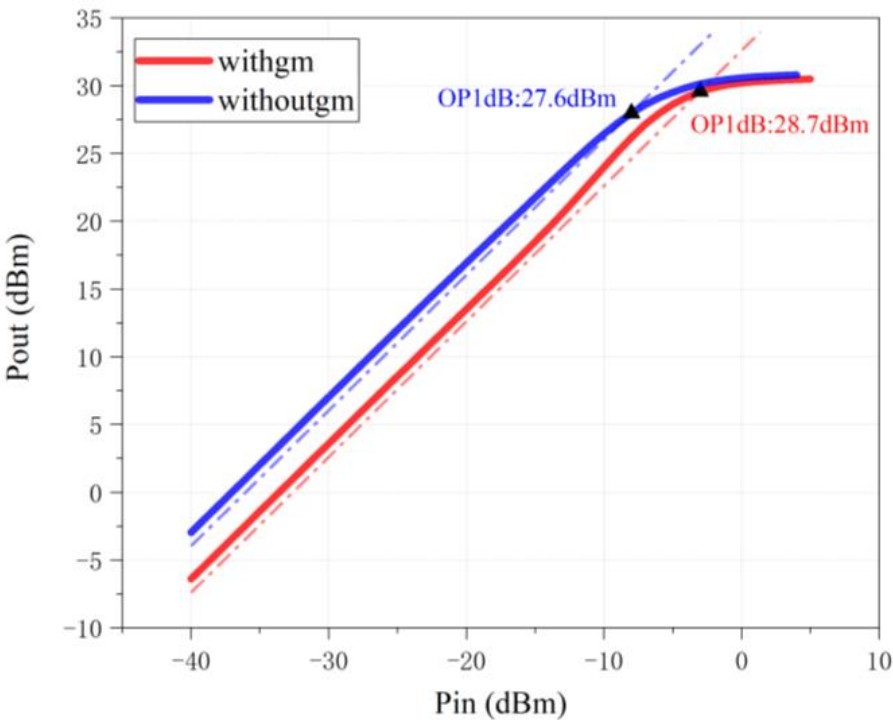

**Figure 8.** Simulated comparison of the output power.

Additionally, the PA's IMD3 also can be improved by gm linearization. To explain this more clearly, we review the regeneration process of the IMD3 [41]. The drain current(id), the most significant non-linear distortion source, can be expanded in the following Taylor series,

$$i_d = g_m V_{gs} + g_d V_{ds} + g_{m2} V_{gs}^2 + g_{d2} V_{ds}^2 + g_{md} V_{gs} V_{ds} + g_{m3} V_{gs}^3 + \cdots$$

The gterms represent the transconductance, drain conductance, and cross terms. $R_L(\omega_L)$ assumes the output load impedance at the fundamental frequency. The input signal $V_{in}$ is given by

$$V_{in} = A[\cos(\omega_1 t) + \cos(\omega_2 t)]$$

Thus, the IMD3 terms can be derived as

$$V_{ds}(2\omega_2 - \omega_1) \approx R_L(\omega_L)[\tfrac{3}{4} g_{m3} A^3 + \tfrac{1}{2} Z_L(\omega_2 - \omega_1) g_{m2} g_{md} A^3 + \tfrac{1}{4} Z_L(2\omega_c) g_{m2} g_{md} A^3$$

$$V_{ds}(2\omega_1 - \omega_2) \approx R_L(\omega_L)[\tfrac{3}{4} g_{m3} A^3 + \tfrac{1}{2} Z_L(\omega_1 - \omega_2) g_{m2} g_{md} A^3 + \tfrac{1}{4} Z_L(2\omega_c) g_{m2} g_{md} A^3$$

where $\omega_c = \frac{(\omega_1 + \omega_2)}{2}$, $Z_L$ is the frequency-dependent load impedance, and $Z_L = R_L(\omega_L)$ at the fundamental frequency.

It can be seen that IMD3 has a great relationship with $g_{m3}$. Figure 9 shows the simulated $g_{m3}$ versus bias voltage. Therefore, low $g_{m3}$ can be realized using a combination of two different bias voltages. Figure 10 shows the comparison of IMD3 between with- and without-gm linearization, indicating better IMD3.

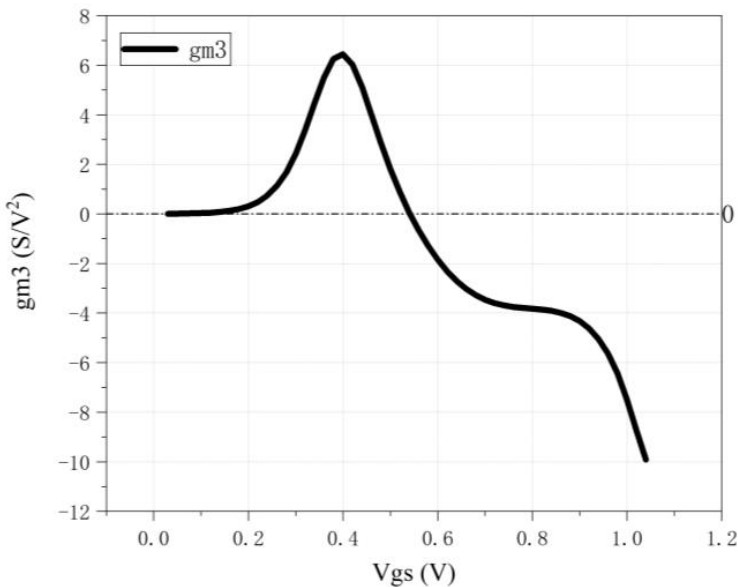

**Figure 9.** Simulated $g_{m3}$ versus bias voltage.

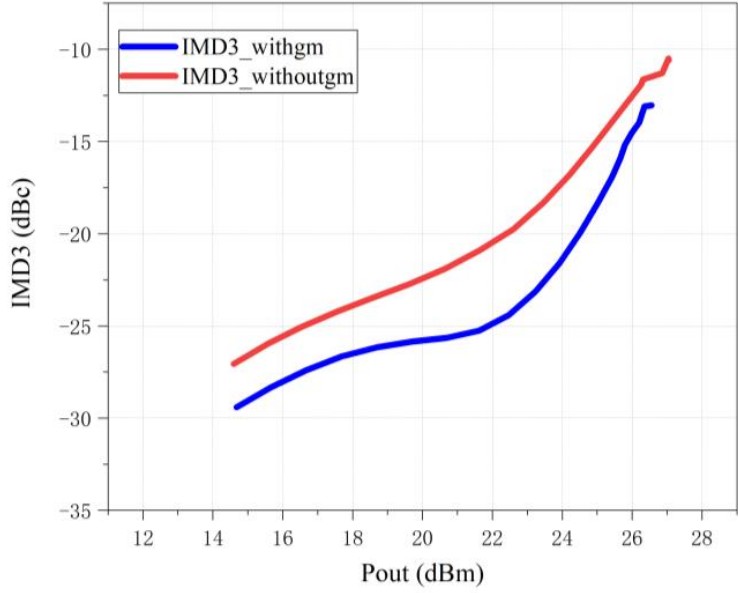

**Figure 10.** Simulated comparison of the IMD3 according to the output power with/without gm.

However, since the power cells are all biased to class AB, when the input signal is large enough, the device will constantly turn on and off, causing the gate capacitance of NMOS Cgg to change dramatically and resulting in severe AM–AM and AM–PM distortion, thereby deteriorating the overall linearity.

In order to compensate for these distortions, a nonlinear capacitance compensation technique is employed using PMOS transistors alongside NMOS transistors in the second stage, since the change of PMOS gate capacitance is opposite to NMOS. Under appropriate size and biasing voltage, the total capacitance seen at the NMOS gate will remain approximately constant, thereby compensating for the distortions.

Figure 11 shows the NMOS gate capacitance, PMOS gate capacitance, and total capacitance. It can be seen that PMOS can compensate for the changes in NMOS gate capacitance in a wider input voltage range, thus achieving high linearity. Figure 12 shows the comparison of IMD3 between with- and without-PMOS compensation, indicating better IMD3.

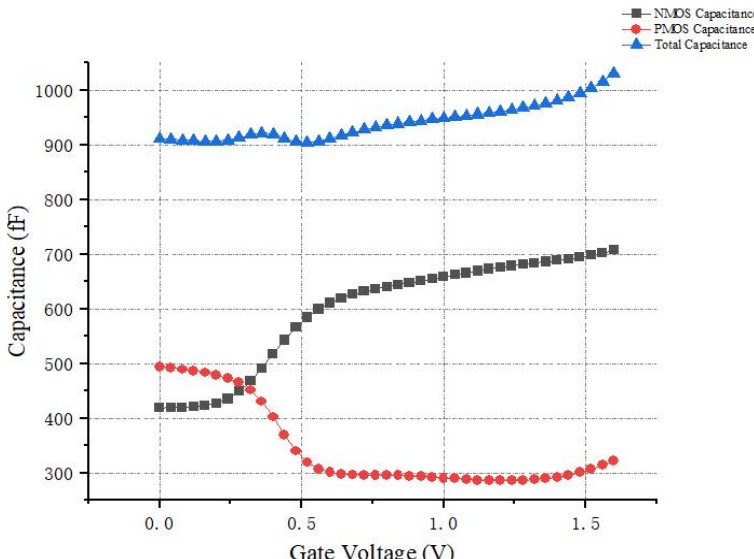

**Figure 11.** Simulated NMOS capacitance, PMOS capacitance, and total capacitance.

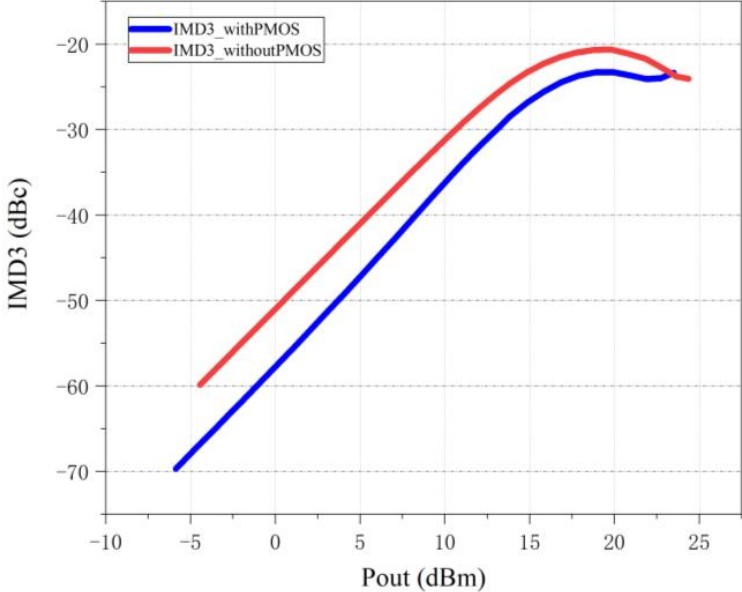

**Figure 12.** Simulated comparison of the IMD3 according to the output power with/without PMOS.

### 3.3. On-Chip Transformer Matching Circuit

The input, interstage, and output matching in this design are accomplished through the utilization of transformers and capacitors. The incorporation of transformers offers several significant advantages. Firstly, transformers inherently provide isolation between the primary and secondary DC paths. Secondly, they exhibit excellent impedance transformation characteristics, thereby enhancing the flexibility of the design. Thirdly, transformers possess wide bandwidth capabilities. Lastly, transformers can be employed as baluns, enabling conversion between single-ended and differential signals.

The Z-parameter matrix of a practical transformer can be represented as follows (The orientation of primary current and secondary current is consistent with Figure 13):

$$Z = \begin{bmatrix} R_1 + j\omega L_1 & -j\omega M \\ j\omega M & -(R_2 + j\omega L_2) \end{bmatrix} \quad M = k\sqrt{L_1 L_2}$$

where k represents the coupling coefficient between the primary and secondary coils in the transformer. When the secondary coil is connected to the load, the equivalent circuit diagram can be represented as follows:

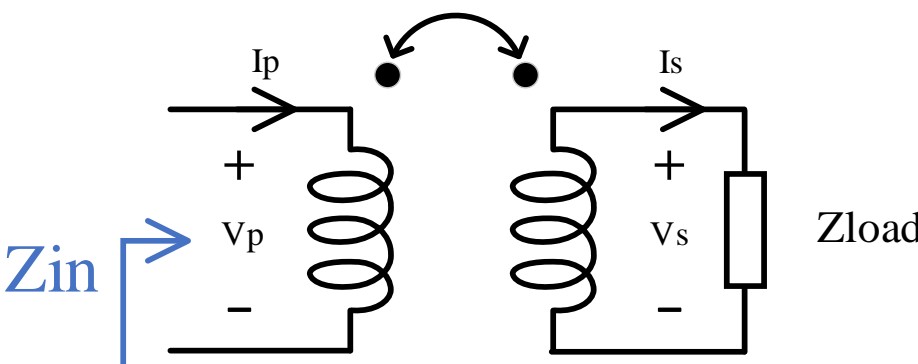

**Figure 13.** Transformer equivalent circuit.

Consequently, the load resulting from the conversion after the primary coil can be obtained as follows:

$$Z_{in} = \left\{ R_1 + \frac{(\omega M)^2 (R_2 + R_L)}{(R_2 + R_L)^2 + (\omega L_2 + X_L)^2} \right\} + j \left\{ \omega L_1 - \frac{(\omega M)^2 (\omega L_2 + X_L)}{(R_2 + R_L)^2 + (\omega L_2 + X_L)^2} \right\}$$

$$Z_{load} = R_L + jX_L,$$

$$M = k\sqrt{L_1 L_2}$$

where k represents the coupling coefficient between the primary and secondary coils in the transformer.

By selecting different values for the inductance and the coupling coefficient (k), it is possible to achieve various impedance transformations, thereby providing a significantly high degree of design flexibility.

However, part of the reason why integrating a power amplifier is challenging is due to the low-Q of the on-chip transformer or inductor, which has a significant impact on the overall linearity and efficiency of the PA. To address this issue, this design adopts the following techniques for on-chip transformer design: First, both the primary and secondary coils are made of the top metal layer, which has the lowest sheet resistance and highest current capacity. Second, a poly shielding is used underneath the coils to reduce the impact of the substrate. Third, to prevent induced currents in the substrate due to electromagnetic fields, a low doping concentration substrate is used beneath the transformer coils. Finally, this work superimposes the AL RDL layer on the primary coil, which slightly increases the Q value of the primary coil. Figure 14 shows the layout of the output transformer and Table 2 shows EM-simulated transformer parameters.

**Table 2.** Simulated transformer parameters.

| Lp | 526.69 pH | Ls | 1.68 nH |
|---|---|---|---|
| Qp | 10.96 | Qs | 15.22 |
| k | 0.726 | | |
| SRF | >10 G | | |

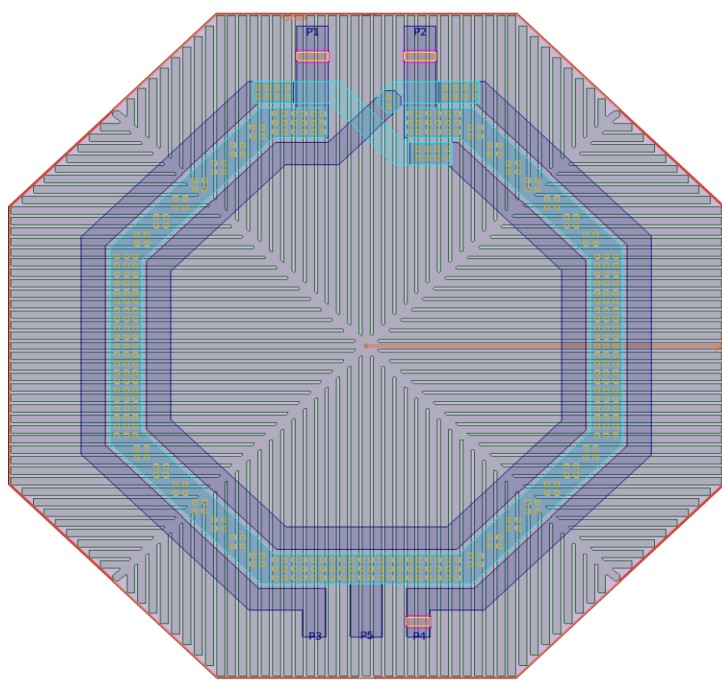

**Figure 14.** Output Balun layout.

Meanwhile, in order to maintain consistent power gain among different channels, the inter-stage matching capacitance is reconfigurable in this design. Figure 15 depicts the schematic of the reconfigurable inter-stage matching capacitance, while Table 3 lists the various capacitance values corresponding to different cap codes.

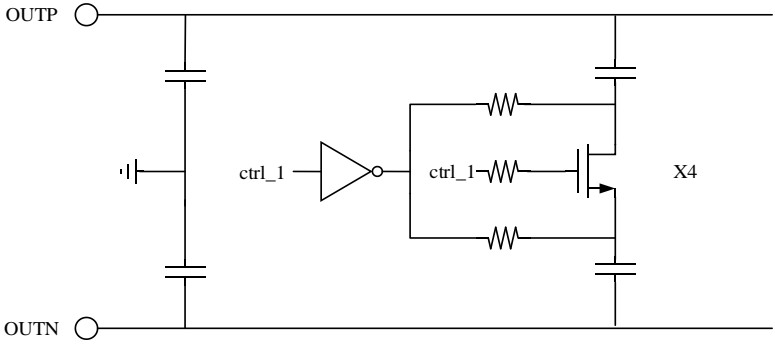

**Figure 15.** Schematic of the Capbank.

**Table 3.** Simulated Capacitance.

| CapCode | Cap | Q |
|---------|-----|---|
| 4′b0000 | 163.4 fF | 28.8 |
| 4′b0001 | 193.18 fF | 33.7 |
| 4′b0011 | 262.68 fF | 35.5 |
| 4′b0111 | 401.76 fF | 33.6 |
| 4′b1111 | 680.02 fF | 30.6 |

*3.4. Integrated Bias Circuits*

The linearity of a power amplifier is closely associated with its biasing, and the integration of bias circuits in the CMOS process can be leveraged to optimize the linearity of the power amplifier, which represents one of the advantages of the CMOS process. In the literature, reference [6] used a common-source bias circuit to inject the envelope signal, effectively mitigating the generation of IMD3 signals within the circuit and significantly

enhancing the IMD3's performance, particularly under conditions of high signal output power. Similarly, reference [41] utilized a dynamic bias control circuit to regulate the common-gate bias voltage at different power levels, thus achieving linearity optimization of the circuit. This research demonstrates that integrated bias circuits can be effectively employed to optimize the overall linearity of the circuit.

The integrated bias circuit utilized in this design is depicted in the figure below (Figure 11). Figure 16a shows the common-source bias circuit, which employs a specific circuit structure for generating the desired voltage. As illustrated in the Figure 16, a variable current source passes through the diode-connected NMOS to produce the voltage, with the current being generated by the BandGap circuit. The proportional to absolute temperature (PTAT) current is utilized to compensate for variations in the threshold voltage caused by temperature changes. The relationship between the threshold voltage of the common-source transistor and the temperature variation is demonstrated in Figure 17. To maintain the stability of the bias voltage, the generated voltage is then fed into a voltage buffer. This voltage buffer prevents any fluctuations in the bias voltage, which could otherwise impact the linearity of the power amplifier. Moreover, it ensures the stability of the output impedance of the bias circuit. The gain-bandwidth product (GBW) of the voltage buffer circuit is also meticulously designed to suppress the RF frequency while ensuring the DC voltage output.

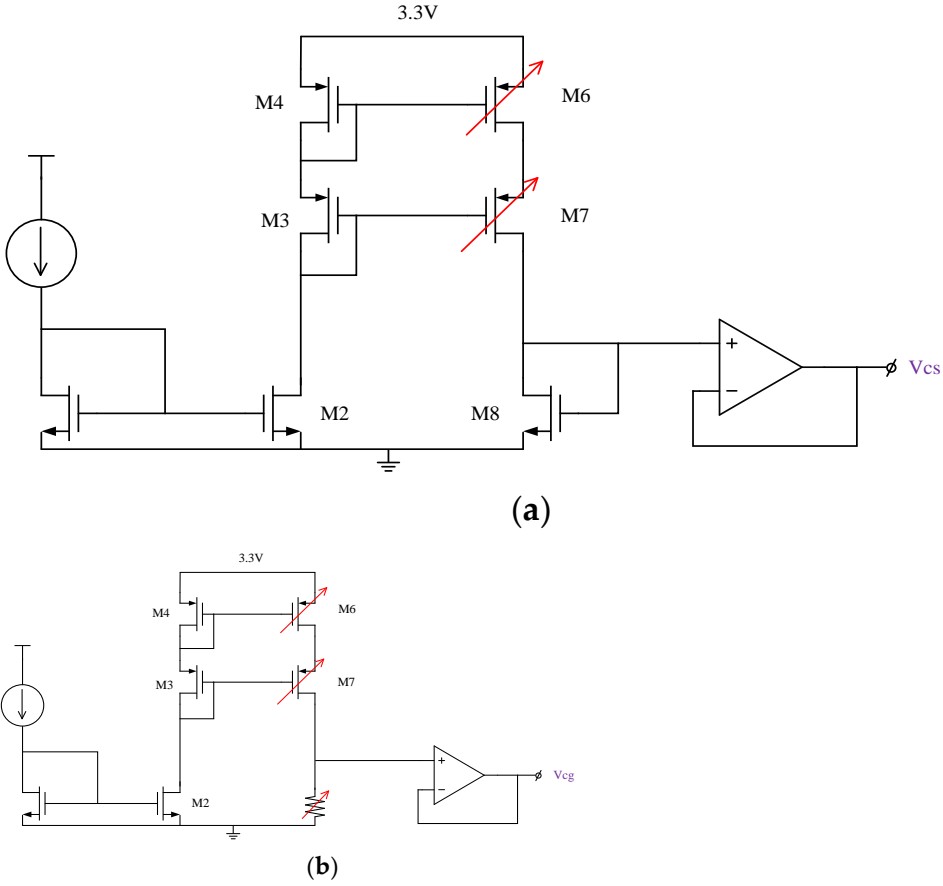

**Figure 16.** (**a**) Schematic of the common-source bias circuit. (**b**) Schematic of the common-gate bias circuit.

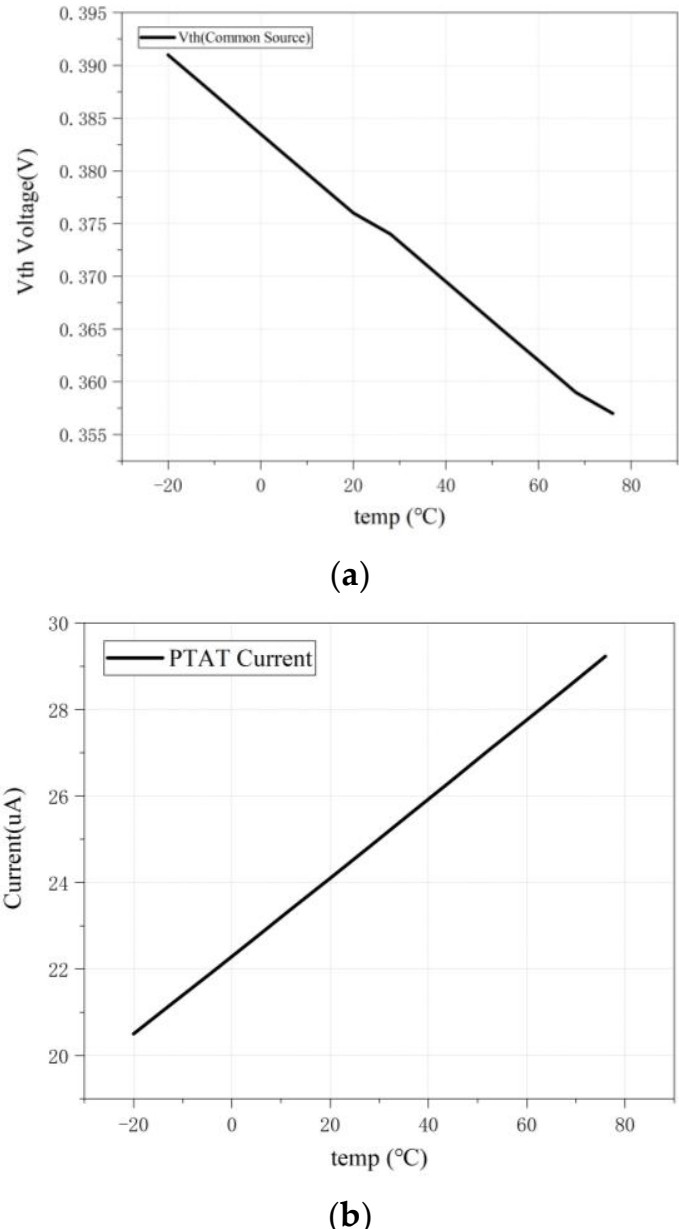

**Figure 17.** (**a**) Simulated threshold voltage versus temperature. (**b**) Simulated PTAT current versus temperature.

The schematic of common-gate bias circuit, illustrated in the Figure 16b, utilizes a variable resistor to generate the required voltage. Similarly, the output of this circuit is connected to a voltage buffer to stabilize the output impedance of the bias circuit.

The simulated output impedance of the bias circuit, as displayed in the Figure 18a, indicates that the addition of the voltage buffer results in a consistent output impedance at different bias voltages. Figure 18b shows that bias circuit impedance without a voltage buffer has a large variation at different output dc voltages, which will affect the linearity of the power amplifier (PA).

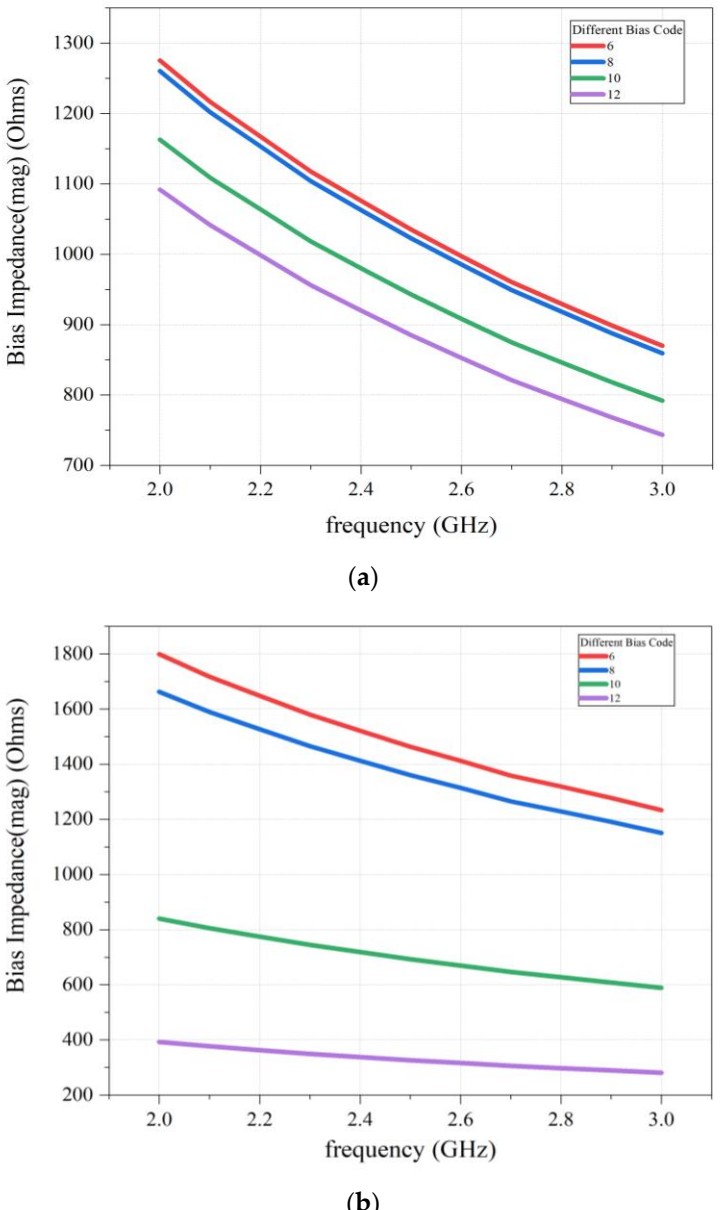

**Figure 18.** (**a**) Simulated bias impedance with voltage buffer at different bias voltage. (**b**) Simulated bias impedance without voltage buffer at different bias voltage.

## 4. Measurement Results

The proposed PA has been implemented in 55-nm CMOS, occupying an area of 1.37 mm × 0.66 mm, as shown in Figure 19. Multiple grounding pads are placed to reduce the inductance of the bonding wires. For the measurements, an off-chip LC balun with a 50–50 ohm turn ratio is used at the PA's input to implement the single-ended to differential conversion. All the measurements are conducted at the output of the PA, and the balun loss has been included in the following measured results. Figure 20 shows the large signal CW measurements' setup and the measurements' environments.

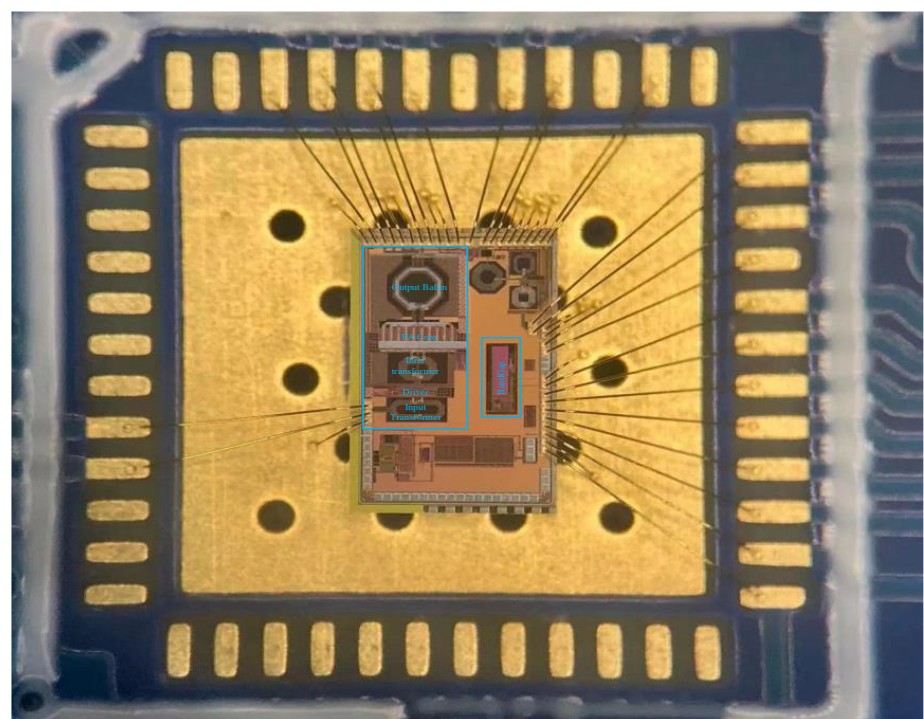

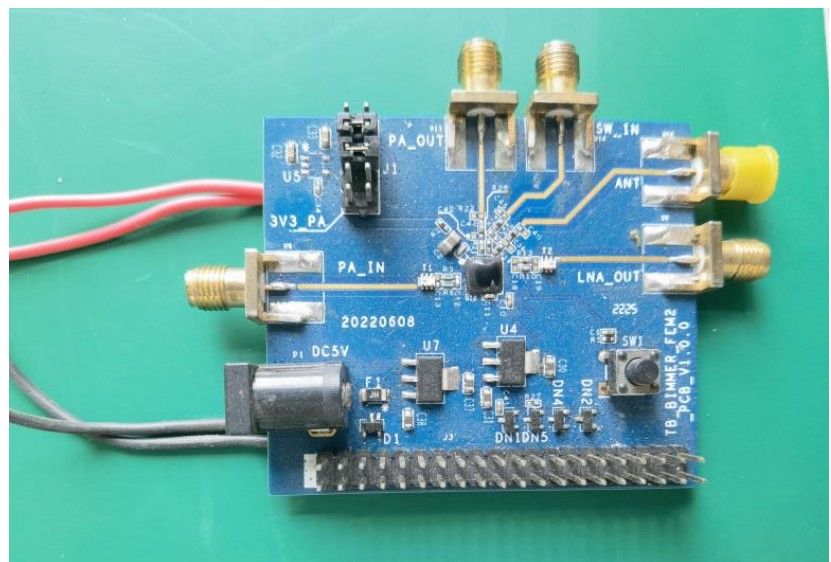

**Figure 19.** Die photo and PCB photo.

Figure 21 shows the measured small signal gain characteristics in pure HP and LP mode by the static control signal.

Figure 22 presents the measured single tone (continuous wave, CW) performance of the PA at pure LP and HP modes, demonstrating a significant improvement in PAE and DC current in the pure LP mode. The PA achieves an output P1dB of 27.6 dBm in pure HP mode with 32.7% max PAE, and 17.7 dBm in pure LP mode with 10% PAE. Notably, in the output power range of 10~15 dBm, the PAE in pure LP mode is 1.5 times that of the pure HP mode. Moreover, the total AM–AM and AM–PM distortions are very low in both modes.

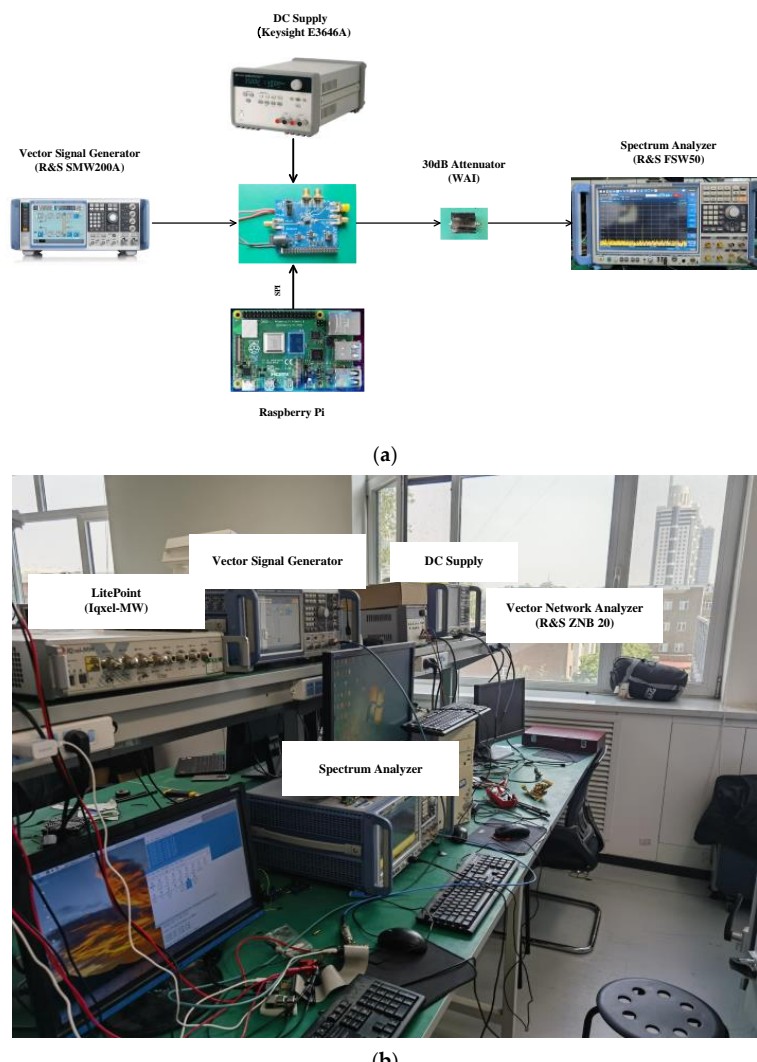

(**a**)

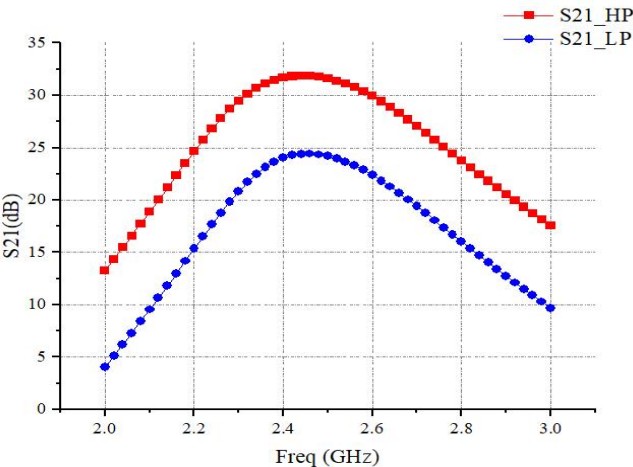

(**b**)

**Figure 20.** (**a**) Large-signal CW measurements' setup. (**b**) Measurements' environment.

**Figure 21.** Measured S21 small signal gain.

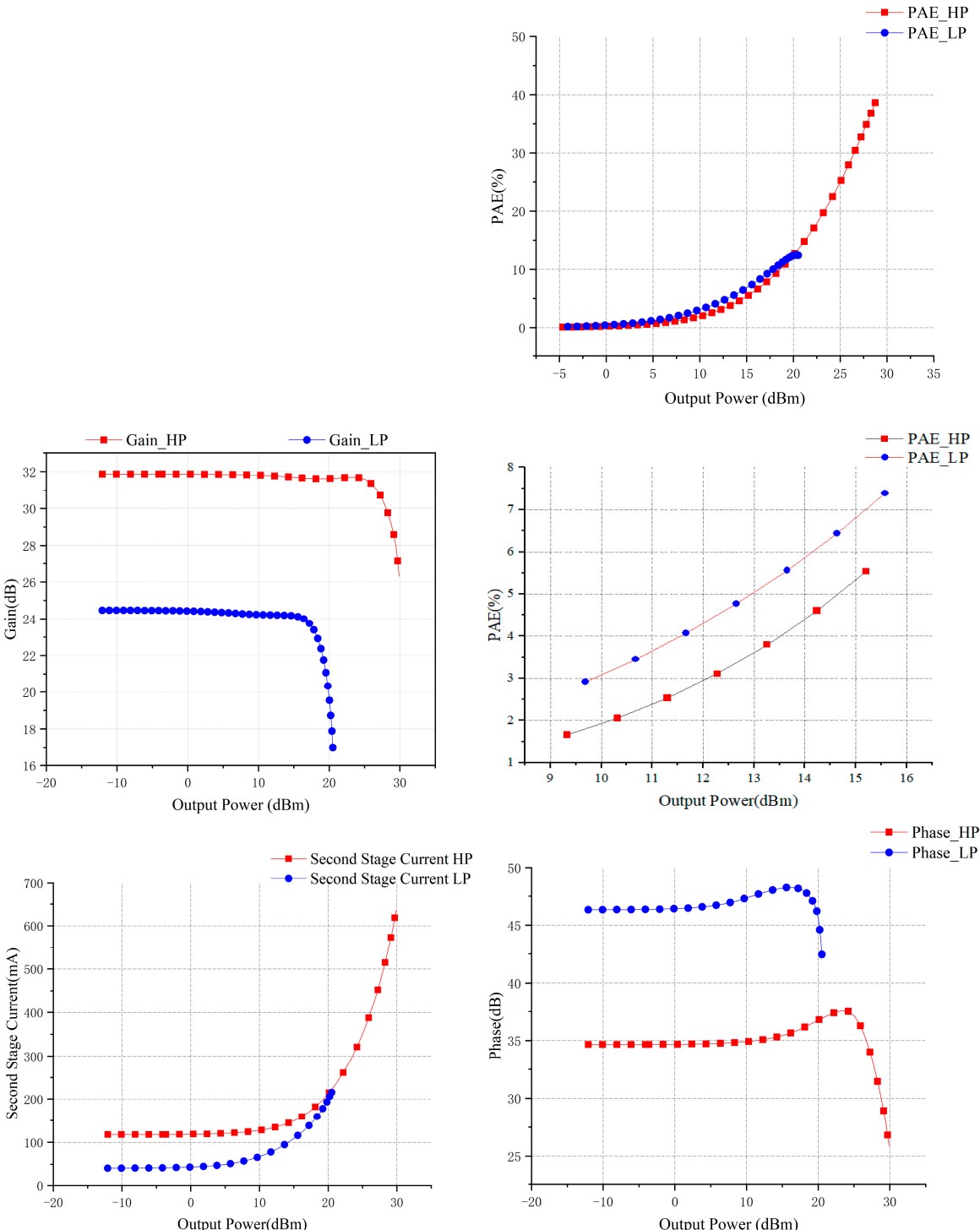

**Figure 22.** Measured CW performance in pure HP and LP mode.

Figure 23 presents the measured single tone (continuous wave, CW) performance of the PA at the pure LP and HP modes with the same gain to ensure that the power control mode works properly. The results show that in this case, there is still a large improvement in PAE, and the gain difference is within 1 dB.

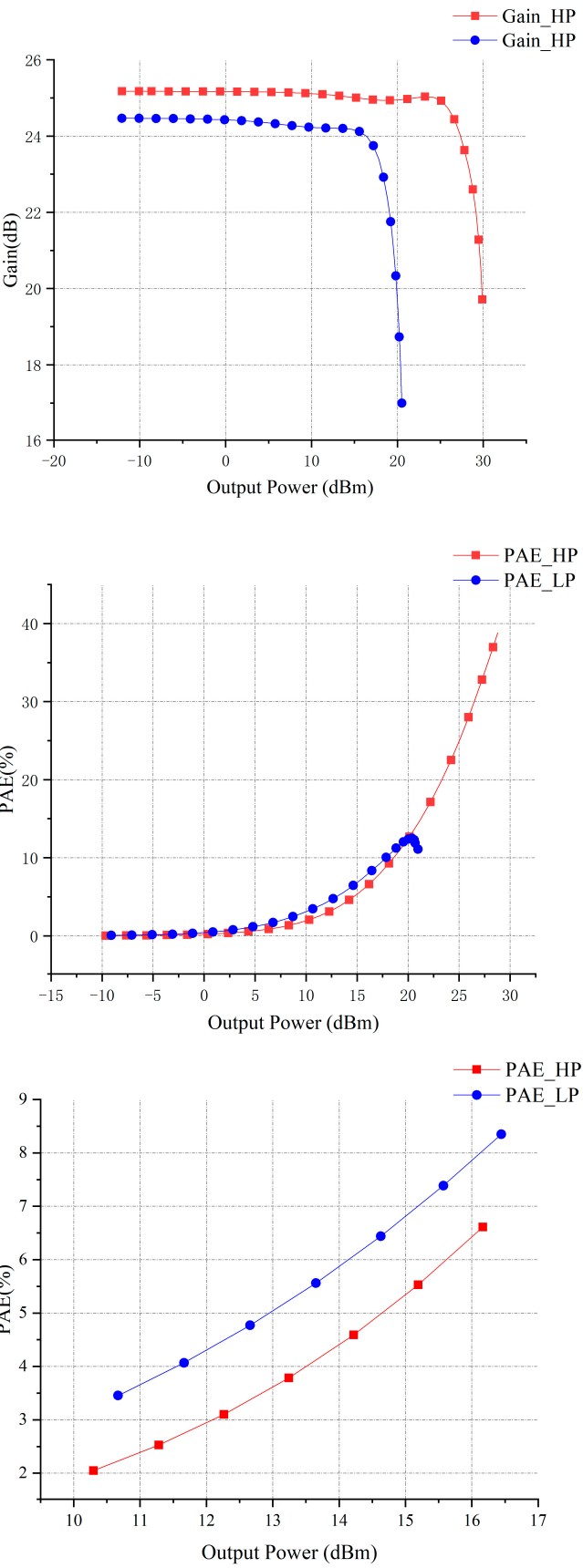

**Figure 23.** Measured CW performance with same gain.

Figure 24 presents the measured CW performance of the PA in power control mode. It can be seen that the DC current is significantly reduced by about 100 mA at low power levels, and the PAE is also improved while maintaining acceptable gain and phase discontinuities.

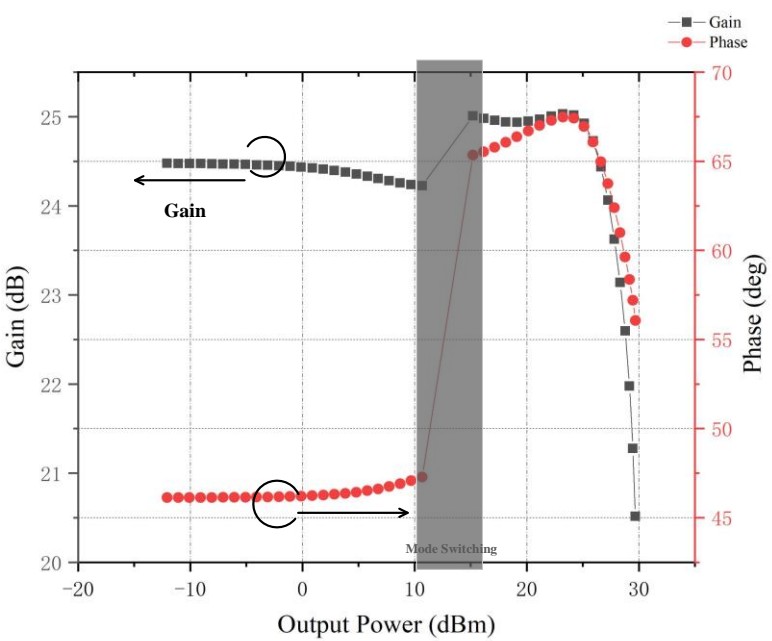

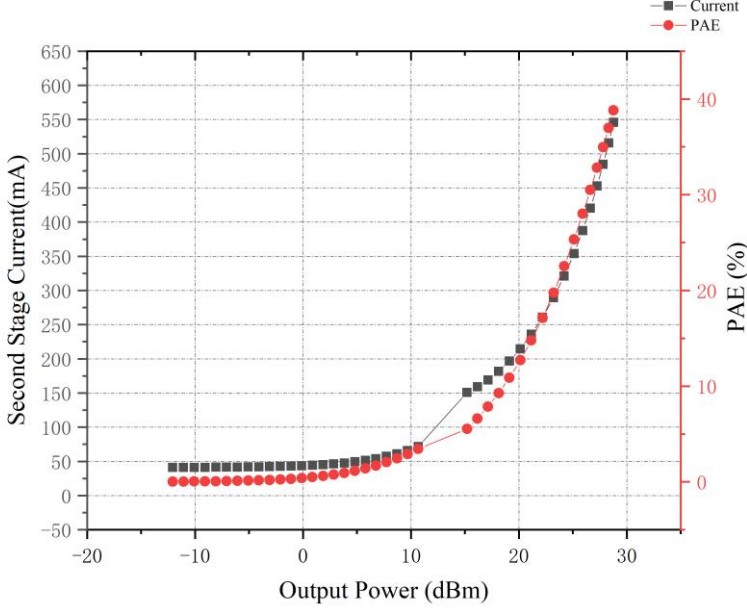

**Figure 24.** Measured CW Performance in power control mode.

Finally, to verify whether the linearity characteristics of the proposed PA are suitable for WLAN system applications, modulation signals are applied. Figure 25 shows the measured performance for an 802.11 n MCS7 signal. The results indicate that even without digital pre-distortion (DPD), the proposed PA meets the linearity requirements of the WLAN 802.11n protocol.

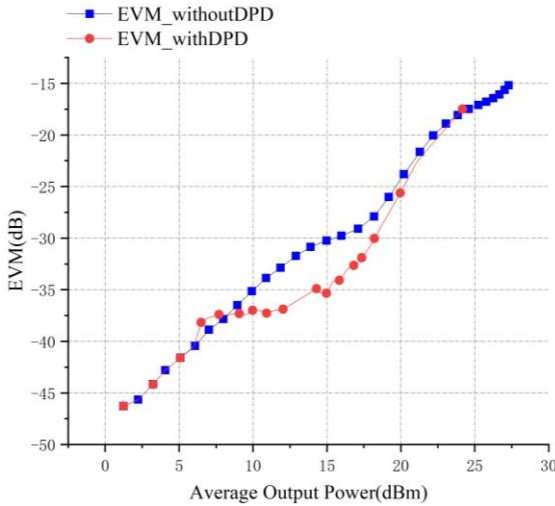

**Figure 25.** Measured PA performance for 802.11n.

Table 4 shows a comprehensive summary of the performance of the presented CMOS PA and compares it with other similar works. In comparison to the state-of-the-art designs, the proposed PA achieves a similar or better performance with a much smaller chip area.

**Table 4.** Comparison with previous works.

| Ref | Freq | Supply | Process | Method | HPM | | LPM | | CW PAE Improve | Linearity |
|-----|------|--------|---------|--------|-----|-----|-----|-----|----------------|-----------|
| | | | | | OP1dB | PAE | OP1dB | PAE | | |
| [40] | 2.4G | 5.6 V | 180 nm-CMOS | Mode Switching | 27 | 26.1 | 22 | 21.8 | x2 | EVM:-15.8@26dBm power, 802.11n |
| [42] | 2.4G | 3.3 V | 90 nm-CMOS | Turn-off one-stage | 22.7 | 12.4 | 18 | 10.4 | x1.6 | - |
| [43] | 2.4G | 3.3 V | 55 nm-CMOS | Dual Mode with Linearizing Body Network | 22 * | 14 * | 21 * | 11 * | - | EVM:-27dB @17.3dBm power, 802.11n |
| [44] | 2.4G | 3.3 V | 180 nm-CMOS | Dual Mode with Tunable Matching Network | 22.3 | 40 | 16.0 | 28 | - | EVM:-25dB @15.7dBm power, 802.11n |
| [7] | 2.5G | 3.3 V | 180-nm CMOS | Discrete resize and PCT | 27.5 | 24 | 20.5 | 15 | x1.5 | - |
| This work | 2.4G | 3.3 V | 55-nm CMOS | Mode Switching | 27.6 | 32.7 | 17.7 | 10 | x1.5 | EVM:-27dB @ 19dBm power, 802.11n |

* Graphically estimated.

## 5. Discussions

With the continuous advancement of WLAN technology, the demand for wider bandwidth (320 MHz or higher) and more intricate modulation schemes (4096 QAM or higher) has emerged to achieve greater data throughput. Within the transmitter, the power amplifier (PA) plays a critical role in determining the efficiency and linearity of the entire transmission path. To satisfy the system's error vector magnitude (EVM) requirements, the power amplifier necessitates exceptionally high linearity and efficiency. Consequently, the design of CMOS PAs faces increasingly demanding challenges.

Looking ahead, the future research direction of CMOS PAs for WLAN primarily revolves around the exploration of high linearization techniques and the enhancement of deep back-off efficiency. These areas of focus aim to address the need for improved linearity performance and increased efficiency, aligning with the evolving requirements of WLAN systems.

## 6. Conclusions

This paper has presented a fully integrated linear 2.4-GHz CMOS power amplifier with two operating modes for WLAN applications. The mode switching is achieved by statically or dynamically controlling the number of power cells. The PA achieves an output P1dB of 27.6 dBm and 17.7 dBm with a power-added efficiency (PAE) of 32.7% and 17.7% in the high-power mode (HPM) and low-power mode (LPM), respectively. Additionally, the efficiency of the low-power mode is 1.5 times higher than that of the high-power mode within the output power range of 10–16 dBm. In the high-power mode, the use of gm linearization and PMOS compensation result in an output 1 dB compression point of 27.6 dBm and a saturated output power of 29.7 dBm. Furthermore, under a WLAN 802.11n 64 QAM modulation signal, the power amplifier achieves −27 dB EVM at 19 dBm output power, demonstrating its ability to meet the linearity requirements of WLAN standards.

The proposed PA could be used in various termini, including routers and tablets. The future direction is to integrate this PA into an entire transceiver to realize a fully integrated WLAN SOC chip. Moreover, the proposed PA has only two modes to switch, and more modes can be achieved in the future to further improve efficiency.

**Author Contributions:** Conceptualization, H.S., T.M. and B.W.; Methodology, H.S., T.M. and B.W.; Software, H.S., T.M. and B.W.; Validation, H.S. and T.M.; Formal analysis, H.S. and B.W.; Investigation, H.S. and T.M.; Resources, H.S., T.M. and B.W.; Data curation, H.S.; Writing – original draft, H.S.; Writing – review & editing, H.S., T.M. and B.W.; Visualization, H.S.; Supervision, H.S. and T.M.; Project administration, H.S., T.M. and B.W.; Funding acquisition, T.M. and B.W. All authors have read and agreed to the published version of the manuscript.

**Funding:** This research received no external funding.

**Institutional Review Board Statement:** Not applicable.

**Informed Consent Statement:** Not applicable.

**Data Availability Statement:** The data in this paper are all from simulation and test results of real circuits, and there is no data plagiarism or falsification. Due to project confidentiality requirements, we regret that we cannot release more details of the data at this time.

**Conflicts of Interest:** The authors declare no conflict of interest.

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
