# Peer review of "A Fully Integrated High Efficiency 2.4 GHz CMOS Power Amplifier with Mode Switching Scheme for WLAN Applications"

_applsci, doi:10.3390/app13137410_

Round 1

Reviewer 1 Report

Both simulation and measurement results are discussed. Paper seems good.However, there are some suggestions as:

1.    The advantage of the proposed design is not clear.

2.    Important characteristic and contribution must be clearly stated in the conclusion

3.       Bandwidth improved configurations were suggested in [ref] need to add in the introduction section.

4.       What are other feasible alternatives? What are the advantages of adopting this technique over others in this case? How will this affect the results? The authors should provide more details on this.

5.       Some assumptions are stated in various sections. Justifications should be provided on these assumptions. Evaluation on how they will affect the results should be made.

6.       The operating mechanism is not well explained.

7.    Literature must include few recent advancements of WLAN applications like:

1. Kumar A, Imaculate Rosaline S. Hybrid half-mode SIW cavity-backed diplex antenna for on-body transceiver applications. Applied Physics A. 2021 Nov;127(11):1-7.  doi:10.1007/s00339-021-04978-9

2. Chaturvedi D, Raghavan S. Dual-Band, Dual-Fed Self-Diplexing Antenna. In 2019 13th European Conference on Antennas and Propagation (EuCAP) 2019 Mar 31 (pp. 1-5).

----------

Author Response

Thank you for your valuable comments and suggestions.Here are my reply

Reviewer 2 Report

The article contains only 3 equations, which in my opinion is quite few for a scientific article. These equations should be numbered and referenced in the text. A comment must be added to the equation for Z on line 208 that it applies to the orientation of the currents according to Fig.8. If the generally used sme of the current I2 were considered, the second row of the matrix for Z would have to have opposite signs (i.e. negative for the first term and positive for the second).

I consider it a positive contribution that the proposed amplifier was not only simulated but also practically realized. Furthermore, the results of the simulations are accompanied by the results of the measured values on the realized power amplifier (PA). However, in the article I am missing the layout of the measuring workplace, i.e. its block diagram, possibly supplemented with its photo. I suggest adding them, it will add clarity to the article. It would also be appropriate to state with which instruments the measurement was made (specific type of measuring device, or its accuracy).

The article is intended for dedicated readers - experts. It assumes knowledge of the definitions of basic measured parameters (e.g.: s21 etc.), therefore it will be somewhat incomprehensible to the uninitiated in the field.

Author Response

(The authors gave the same response as above.)

Round 2

Reviewer 1 Report

Revused paper is suitable for publication 

Ok